# Transfer of training—Virtual reality training with augmented multisensory cues improves user experience during training and task performance in the real world

Natalia Cooper[1], Ferdinando Millela[2], Iain Cant[3], Mark D. White[4], Georg Meyer[5]*

1 Construction Research Centre, National Research Council Canada, Ottawa, Ontario, Canada, 2 UKAEA, Culham Science Centre, Abingdon, United Kingdom, 3 Virtual Engineering Centre, University of Liverpool, Liverpool, Merseyside, United Kingdom, 4 Department of Engineering, University of Liverpool, Liverpool, Merseyside, United Kingdom, 5 Department of Psychology, University of Liverpool, Liverpool, Merseyside, United Kingdom

* georg@liv.ac.uk

**Data Availability Statement:** All relevant data are within the paper and its Supporting Information files.

## Abstract

Virtual reality (VR) can create safe, cost-effective, and engaging learning environments. It is commonly assumed that improvements in simulation fidelity lead to better learning outcomes. Some aspects of real environments, for example vestibular or haptic cues, are difficult to recreate in VR, but VR offers a wealth of opportunities to provide additional sensory cues in arbitrary modalities that provide task relevant information. The aim of this study was to investigate whether these cues improve user experience and learning outcomes, and, specifically, whether learning using augmented sensory cues translates into performance improvements in real environments. Participants were randomly allocated into three matched groups: Group 1 (control) was asked to perform a real tyre change only. The remaining two groups were trained in VR before performance was evaluated on the same, real tyre change task. Group 2 was trained using a conventional VR system, while Group 3 was trained in VR with augmented, task relevant, multisensory cues. Objective performance, time to completion and error number, subjective ratings of presence, perceived workload, and discomfort were recorded. The results show that both VR training paradigms improved performance for the real task. Providing additional, task-relevant cues during VR training resulted in higher objective performance during the real task. We propose a novel method to quantify the relative performance gains between training paradigms that estimates the relative gain in terms of training time. Systematic differences in subjective ratings that show comparable workload ratings, higher presence ratings and lower discomfort ratings, mirroring objective performance measures, were also observed. These findings further support the use of augmented multisensory cues in VR environments as an efficient method to enhance performance, user experience and, critically, the transfer of training from virtual to real environment scenarios.

**Funding:** The study was funded by ESRC Case award (North West DTC ES/J500094/1), funder website http://www.esrc.ac.uk/. The findings and conclusions of this article are those of the authors and do not necessarily represent the views of ESRC Research Council. The funders had no role in study design, data collection and analysis, decision to publish, or preparation of the manuscript.

**Competing interests:** The authors have declared that no competing interests exist.

# Introduction

Learning in virtual reality (VR) is a compelling solution for situations where training is difficult in real life, for example because the consequences of errors would be too grave for the trainees (e.g. flight simulation) or others (e.g. medical education). Other application areas include training scenarios where training exercises would be either impracticable (emergency drills) or too costly to be practised in reality.

There is no question that skills learnt in VR significantly reduce error rates and successfully transfer into a real settings, such as clinical [1–3] or flight training [4]. Cook and colleagues [5] systematically reviewed 609 studies with 35226 medical trainees and reported that medical simulation training is consistently associated with large effect sizes for outcomes in the areas of knowledge, skills, and behaviours, and moderate effects for patient-related outcomes. Corresponding findings have been shown in a meta-analysis of flight simulation training for pilots that shows consistently better outcomes compared to aircraft training only [4].

There is therefore a convincing general case for VR-based training, but an open question is how to optimise training in VR. Systematic reviews report significant variability in outcomes that can be attributed to specific training regimes, tasks, and equipment used. An additional fundamental question is whether subjective or objective performance gains, seen during VR training, will transfer into real life situations [6]. Therefore, further investigation of the main factors that can aid effective learning transfer is needed. We have previously shown that additional, task relevant cues can improve performance and enhance users' acceptability of VR simulation [7] even if these cues reduce the overall fidelity of the virtual environment. In this study, we investigate whether VR training with augmented multisensory cues in virtual environments translates into performance improvements in real environments.

## The main factors affecting VR experience

There are many factors that can affect training and learning in VR. Multisensory feedback during the interaction in VR has been shown to be one of the main factors. Most VR scenarios rely on the visual modality as the main source of sensory stimulation. Audio and haptic signals, however, have also been shown to facilitate performance in virtual environments. Previous studies have shown that multisensory signalling can be very effective in dual manipulation tasks [8], surgical simulation [9], rehabilitation [10] and flight training [11].

It is well understood that performance in VR training environments is directly affected by simulation fidelity [11–13]. Poor mechanical performance of simulated haptic feedback, for example, has been shown to result in negative training outcomes in surgical training [14]. Conversely, improved simulation fidelity not only captures the sensory cues and behaviours that learners encounter in real life better, but it also improves presence and immersion in VR environments [11–13,15–17] while reducing cognitive load [18–20]. 'Immersion' here stands for the objective level of sensory fidelity provided by a VR system [15] which is a direct measure of simulation fidelity. 'Presence' describes the subjective psychological response of a user experiencing a VR system as 'being in the virtual world' (see [7, 21] for further discussion of these terms). For successful transfer of training from virtual into real environments, it therefore seems essential that the cues presented during the training in VR are representative of those encountered in reality.

Practical considerations, such as financial and operational constraints, may sometimes limit the degree to which sensory stimuli can be provided in many VR applications [22]. While there is some progress in providing haptic cues [23–26], there still are significant challenges [27] that can ultimately decrease overall task efficiency when haptic cues are presented [28].

Other cues, such as vestibular, olfactory, or gustatory cues are even less well developed for mainstream learning applications.

Learning and user experience in VR does not depend only on sensory factors, or fidelity, but also includes significant actional and symbolic components [17]. In the context of image fidelity, for example, a distinction is made between physical fidelity (veridical stimulation of the sensory system), photo-realism (veridical representation) and functional fidelity (veridical representation of the symbolic information) and it has been shown that functional fidelity, in particular, affects performance in VR [11, 29].

One of the first definitions of the 'degree to which the training devices must duplicate the actual equipment' was the umbrella term 'simulation fidelity' [30]. This definition has significantly expanded, not only to include the training environment, but also to represent the various aspects of 'fidelity' that contribute to simulation quality. Liu et al.[31] differentiate between two principal groups of fidelity descriptors that capture either the 'physical' experience or the 'psychological or cognitive' experience. The first group of fidelity descriptors describe how real the virtual environment looks, sounds, or feels, we use the term 'surface fidelity' here to capture this. The terms commonly used to describe the physical experience are simulation fidelity [32], physical fidelity [30], visual-auditory fidelity [33] or equipment fidelity [34]. The second set of fidelity descriptors, measuring the degree to which the simulation captures cognitive factors are psychological-cognitive fidelity [35], task fidelity [34] or functional fidelity [30]. Introducing augmented cues, here, for example, changing the tyre colour to yellow when it is properly seated, has two opposite effects: fidelity descriptors that measure the 'physical realism' of the simulation reduce because the behaviour of the simulated environment is designed not to replicate the real system. Descriptors or 'informational' fidelity, however, increase because the augmented cues were chosen to represent task-critical information that cannot easily be signalled by conventional cues. The augmented cues, by definition, reduce objective measures of immersion, because they make the simulation less like the real system; they also may reduce subjective factors (presence) because they are clear signals that the simulation is a synthetic construct.

In VR environments, functional information can be represented as arbitrary signals and in arbitrary modalities. In the assembly task used in the experiments described below, for example, it was necessary to fully insert a bolt. In real life, this information (bolt meets resistance) would be signalled by simultaneous changes in haptic (torque on the tool), auditory (tool sound) and visual cues (rotation stops). The haptic (torque) cue is difficult to realise in VR [11, 12, 17, 25, 28] but representing this *functional event* is relatively easy in VR by providing alternative haptic signals. We adopted a novel approach to provide an additional informational value of sensory cues during the virtual training. In our virtual simulation, we provided vibration at the top of the hand instead of torque force, a visual colour change of the bolt and the wheel instead of visible rotation and resistance forces, and increased sound intensity instead of the more complex signal change that would be observed in a real tool. By presenting these unrealistic cues during virtual training the overall realism of the virtual environment was decreased.

We have previously shown that the use of augmented multisensory feedback during virtual training, which increases informational content at a cost to physical fidelity, enhances performance and, perhaps counterintuitively, users' perceived sense of presence in VR [7]. On this basis, we advocate the use of augmented cues in VR environments as an efficient method to not only enhance performance and user experience during VR training but also the transfer of training to real environments. As such, the open question that we aim to address here is whether augmented multisensory cues used during VR training enhance the process of *learning transfer* from virtual to real scenarios.

The main objective of this study is to investigate whether virtual training with augmented multisensory cues translates into performance improvements in real environments. We hypothesize that augmented multisensory cues, presented during VR training, will enhance user performance and overall experience in virtual environment, and improved transfer of training into a real task scenario.

## Methods

### Participants

For this study we recruited 42 students and staff at the University of Liverpool via opportunity sampling. Participants were randomly allocated into three training groups that were matched across age, gender, and experience level. Each group consisted of 8 females and 6 males, with an age range between 17 and 60. This study was approved by the University of Liverpool's Institute of Psychology and Health Sciences Ethics Committee (PSYC-1112–049A). All participants gave written informed consent and reported normal or corrected-to-normal vision and normal hearing.

### Task

The training objective was to teach participants to change a racing car tyre as fast as possible using a pneumatic wrench. Participants had to remove 4 bolts, replace the tyre, and then insert and tighten the bolts back to secure the wheel.

The 'real only' (RO) group consisted of participants who performed the real tyre change only, no VR training was given. This group serves as our control group. The other two groups received virtual training before performing the tyre change on a real car. The 'Normal Cueing' (NC) group received conventional training in a typical VR environment with basic visual information only. A third group received additional augmented Auditory, Tactile and Visual cues, (ATV group) during the entire VR training. For more details about the set up please review Cooper et al. paper [7].

### Apparatus

The virtual training was conducted in the Virtual Engineering Centre (VEC) laboratory at the University of Liverpool. The VR setup consisted of one Active Mode display screen with the width of 2.74m and height of 1.72 m, behind which was one active stereo projector that creates 1920 x 1200 resolution images at a rate of 120 Hz (Fig 1A).

When performing the task in the virtual environment, participants were required to wear wireless LCD shutter glasses that were synchronized with the projectors to provide stereoscopic images. 6 high-spec infrared cameras (4 Bonita 10 and 2 Bonita 3) were used to record and track the objects positions: LCD shutter glasses (for head tracking and point-of-view (POV) adjustment), which enabled a 3D stereo view; haptic gloves and a faithful digital mockup of an impact wrench (weight 1.94kg) (Fig 1C–1E). Position data, computed using VICON Tracker software, were broadcast in real-time across the internal network using a Virtual-Reality Peripheral Network (VRPN) protocol at a rate of 200 Hz and used to update the virtual environment. The wheel change simulation was implemented in Unity and run at an average frame rate of 75 fps. Two stereo speakers positioned next to the projector behind the screen provided audio feedback. The action with left hand was aligned with outputs from left speaker, and the action from the right hand we aligned with outputs from right speaker. Other noises included a drilling sound when the tool was in use and wheel alignment sounds when the wheel was manipulated. Two tactile gloves, with a vibration motor attached to the top of the

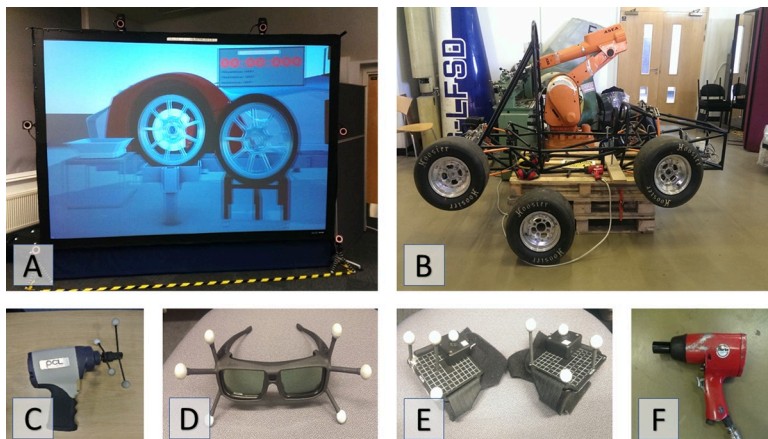

**Fig 1.** Virtual (A) and Real (B) set up for the wheel change. The equipment used in virtual and real tasks: C: mock impact wrench with location markers; D: LCD shutter glasses; E: haptic gloves; F: real impact wrench.

hand, provided tactile stimuli. The vibration occurred with variable frequency ranging from 15 to 250 Hz and variable vibration amplitude (up to 7g). The change of vibration intensity had the purpose of reinforcing learning and providing more detailed information in specific situations. For example, vibration started at an intermediate level when the bolt was first touched, then suddenly increased when the bolt was fully inserted to signal the task-relevant state change. Apart from obvious visual feedback, participants were also provided with additional augmented visual cues, implemented as a colour change (red and yellow) of the simulated hands, wheels, and bolts to signal contact with virtual objects or when critical phases of the task were completed (i.e. wheel in correct position; bolt changes colour when fully inserted).

For the physical wheel-change task, we used the frame of a racing car that was provided by the Formula student team of the University of Liverpool (Fig 1B). The wheels used in the task were attached to the car by four bolts. For the real task, a Clark Air ½" impact wrench (2.1kg), attached to a compressed air line via expandable hose was used (Fig 1F). Participants wore personal protective equipment (PPE) such as protective overalls, steel-toe capped boots, gloves, eye protection and ear defenders.

## Performance measures

The overall time to complete the task was recorded as an objective performance measure. Subjective measures were obtained from questionnaires that participants filled in before, during and after the task. For the VR training groups, the Immersive Tendencies Questionnaire (ITQ) and Presence Questionnaire (PQ) [36] were used to assess participant's immersive tendencies before the study and perceived sense of presence during the interaction in virtual environment; Simulation Sickness Questionnaire (SSQ) was used to assess participants' levels of discomfort during the simulation [37]. The NASA TLX questionnaire [38] was used for the evaluation of cognitive workload experienced during the task in real and virtual environments. All questionnaires can be found in the S1–S3 Questionnaires.

## Procedure

**Real task procedure.** Before the real task, participants filled in pre-test questionnaires and performed a dexterity test to assess their manual motor skills and colour vision. The dexterity

test consisted of organising coloured blocks in a prescribed pattern and screwing bolts to small screws. The experimenter recorded the time when both tasks were finished. Once the dexterity test was completed, the experimenter explained the actual task procedure and showed a full demonstration of the task. Participants were instructed to perform the task as fast as possible and were told that an error penalty of 5 seconds would be applied for any bolt that the experimenter was able to unscrew by hand after the task has finished. The time to complete the task was measured with a stopwatch, starting with the first contact with the wrench and ending when the tool was placed back on the floor. The fastest completion times were displayed on a leader board. After the first trial, participants were required to fill in a workload questionnaire. After this, participants were instructed to perform the task four more times, as fast as possible. Altogether, participants completed the real tyre change task five times. After the final trial, participants filled in another workload questionnaire where they were asked to reflect on the overall task.

**Virtual task procedure.** Before the task, participants filled in pre-test questionnaires to obtain baseline measures and performed a dexterity test to assess their manual motor skills and colour vision. Following this, the experimenter performed a demonstration of the whole task in the VR environment whilst participants observed, wearing 3D glasses. Then participants put on haptic gloves and were instructed to perform the task themselves. The time to complete the task was measured automatically and ranged from the first collision event between hand and wrench to the point when the tool was placed back on a stand. As for the real wheel change task, participants filled in subjective workload as well as sickness questionnaires after the first trial and after completion of the final trial run. After the virtual training was completed, participants were taken into the assembly laboratory where the real wheel change was performed. The procedure for real task was the same as described above.

The only difference between the two VR-trained groups was that the participants in the NC group received training with conventional VR cues (basic visual information but no vibration, no audio and no additional visual information), whilst participants in the ATV group received training with all additional augmented multisensory cues (Auditory, Tactile, and Visual) present at all times during the VR training. The study flow diagram can be seen in Fig 2. The RO group had one practise run on the real tyre change and then performed the task 5 more times where the overall completion times were recorded for data analysis. During the virtual training, the NC and ATV groups had one practice run and then they performed the task 5 more times in VR. When they did the real tyre change, they had one practise run, after which they performed the real tyre change 5 more times where overall completion times were recorded and used for final data analysis.

## Results

### Group comparison

In this study we had three experimental groups. Each group consisted of 14 participants (8 females, 6 males). Mean age for real only (RO) group was 20.6 years (SD = 3.84), for the normal cues (NC) group it was 20.1 years (SD = 2.27) and for the multisensory (ATV) group it was 21.4 years (SD = 0.83). The data from demographic questionnaires showed that there were no significant differences between the groups for age, self-reported previous experiences, or dexterity test scores.

### Objective measures

The time to complete the task was recorded as an objective performance measure. The penalties that participants acquired when the bolts were not fastened in correctly were added to the

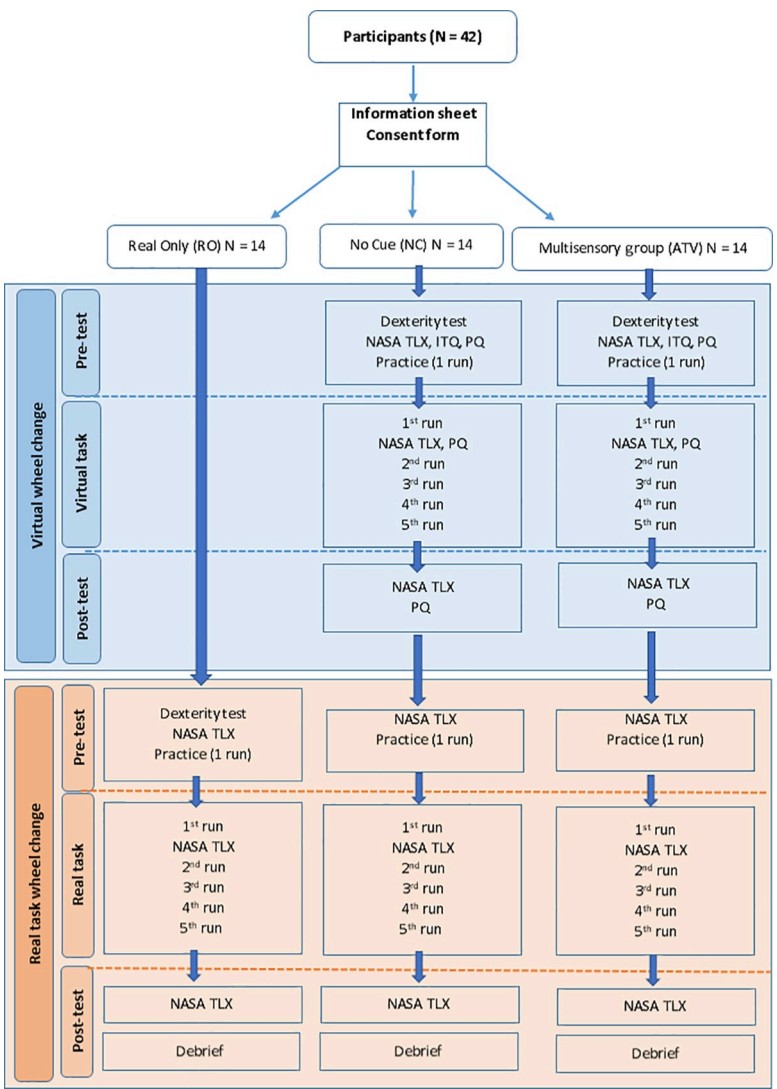

**Fig 2. Study flow diagram.** The diagram shows the procedure during the real and virtual tasks.

overall time and the final analysis was performed on these overall times. Descriptive statistics for overall completion times in all five trials and across all three experimental groups are show in Fig 3.

In order to investigate the performance improvements of the three groups over all trials we conducted a mixed ANOVA test with trial runs (1–5) as a within subject factor and training group (real only—RO, conventional VR training–NC, and augmented cues VR training–ATV) as the between subject factor. Our analysis revealed a significant main effect of trial run (F (8,156) = 22.674, p < 0.001, $n^2$ = 0.37). The interaction between the training groups and trial was not significant (p = 0.081). The main effect of group was also found to be not significant (p = 0.284). These findings suggest that there were significant differences in overall completion times between the first run and the final run, consistent with successful learning.

Independent sample t-tests revealed that at the first trial the completion time between RO and ATC group was significantly different (t = 2.845, p = 0.009) with the ATV group completing the task faster. The NC group also completed the first trial faster than the RO group,

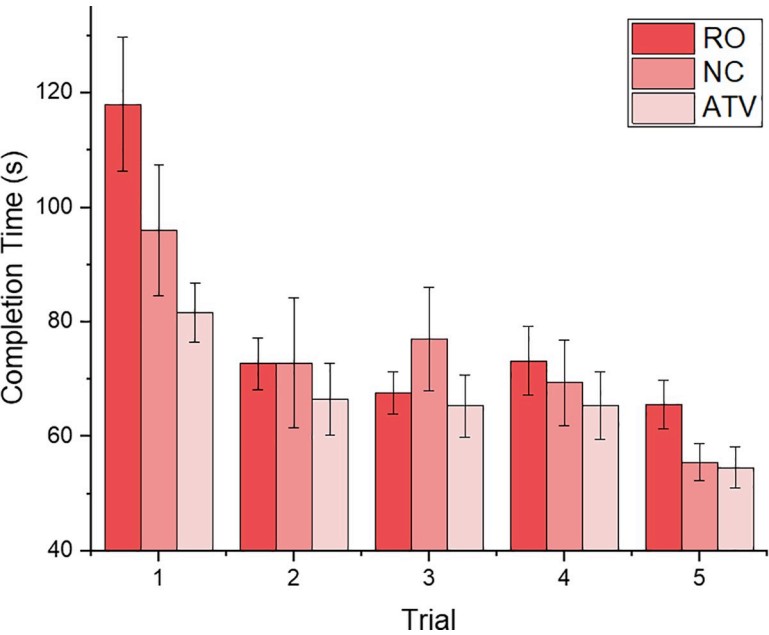

**Fig 3. Mean completion times.** Mean completion times for the real only group (RO red), the minimal cue group (NC dark pink) and the multisensory group (ATV light pink) across all five trials. The error bars represent standard error (SE).

however the difference failed to reach statistical significance (p = 0.198). The overall completion times between the two VR training groups on the first trial were also found to be not statistically significantly different (p = 0.259). On the last trial, the ATV group performed the task fastest when compared to other groups although the overall times were not statistically significantly different from each other (RO group: p = 0.058; NC group: p = 0.847). The difference between the overall times between the RO and NC groups on the last trial also failed to reach statistical significance (p = 0.068).

Thurstone [39] almost exactly 100 years ago observed that"*Learning curves are usually very erratic and for this reason it is necessary to study the general trend of numerous observations instead of the variable individual observations*" and proposed an exponential decay function as a model for learning behaviour. Learning curves are effective tools to quantify and monitor performance of workers exposed to new and repetitive tasks [40] because they parameterise learning behaviour and enable comparisons that go beyond single time points. In order to assess and compare performance between groups, exponential decay functions (Eq 1) were fitted to the performance data for the 'real' tyre change task (Fig 4).

$$y = y_0 + A_1 e^{-\frac{x-x_0}{t_1}} \qquad \text{(Eq 1)}$$

Where $x_0$ and $y_0$ are the x and y offset respectively, $A_1$ is the initial amplitude (time to complete the task) and $t_1$ the time constant defining the exponential decay.

The learning curve, also known as experience curve, represents the improved efficiency or performance obtained from repeating an operation of a specific task by a worker; i.e. the time required to perform a task declines at a decreasing rate as experience with the task increases [41].

To enable a direct comparison of the data the three curves were fitted simultaneously. Parameters $A_1$, $x_1$ and $y_0$ were shared across the three data sets under the assumption that,

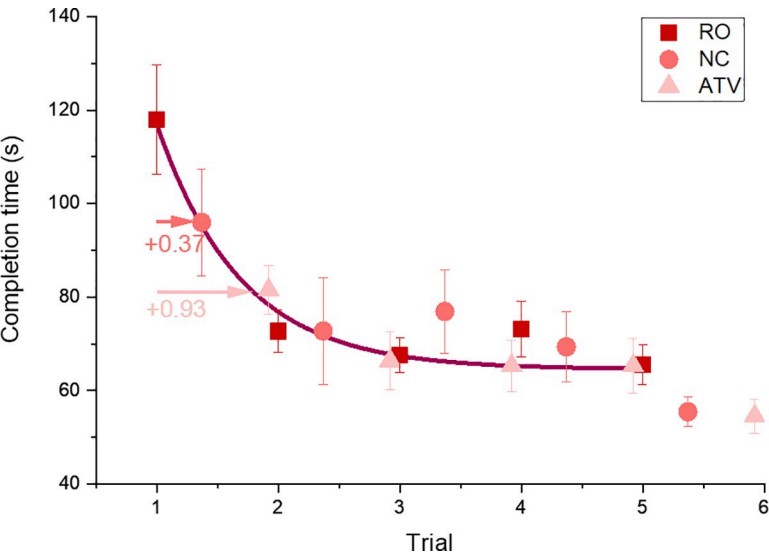

**Fig 4. Mean completion times across three groups in all five trials fitted with exponential curve.** The error bars represent standard error (SE). The X offset indicates a head start for two virtual training groups (RO = real only, NC–No Cues during VR training, ATV—VR training with additional augmented multisensory cues).

since each of the three (matched) groups performed the same task, we assume initial and final performance as well as learning rate to be the same for all three groups. The x-offset ($x_0$), in contrast, was fitted for each individual to quantify the equivalent training time (number of trial runs) that was gained by participating in the VR training exercise preceding the real tyre changes. The difference in $x_0$ between the 'real only' (RO) condition and the two VR training conditions therefore represents the 'real training time saved' by taking part in the initial VR training.

To estimate the shift parameter ($x_0$), a bootstrapping analysis with 10000 resamples (with replacement) was carried out to estimate the mean x-offset ($x_0$) and 95% confidence intervals for each of the three conditions. The mean x-offset ($\bar{x}_0$) for the RO group was 1.10 (CI 0.51–1.63), for the NC group $\bar{x}_0 = 0.554$ (CI -0.0423–1.171) while for the ATV group $\bar{x}_0 = 0.1735$ (CI -0.3903–0.6853). The probability that the observed mean x-offset values were drawn from the RO data are p = 0.0012 (ATV) and p = 0.0329 (NC) respectively. The relative differences in the $x_0$ parameter mean that conventional VR training (NC) is equivalent to 0.37 'training sessions' on a real tyre while the augmented (ATV) VR training provides gains that are equivalent to 0.93 real tyre changes (Fig 4).

Expressing the training outcome in 'equivalent real training sessions' enables a more meaningful and task independent comparison of VR training transfer than a direct comparison of raw performance, such as completion time. It is worth noting that the equivalent gains of NC and ATV VR training represent improvements in training that are 1/3 and 2/3 of the total improvement seen over five real sessions. These results provide strong support for the use of virtual reality as a training platform for task performed in the real environment.

## Improving informational fidelity with augmented feedback

We hypothesised that the additional, information-bearing sensory cues that were presented in the ATV condition would enhance learning outcomes where key events are signalled–in particularly when each bolt is fully tightened. This event was signalled by additional visual (colour) change (in VR the bolt changed colour from yellow to red when it was fully inserted),

intensity increase in haptic (in VR, an increase of vibration was felt in participant's hands when the bolt was fully inserted (when it turned red)) and auditory cues (in VR, participants heard an enhanced thudding sound when the bolt was fully inserted, signaling that the wrench is skipping as it cannot turn the bolt any tighter (as it would during the bolt tightening)). This event was meant to teach participants to attend to these cues signalling when the bolt is fully inserted. Our objective was to see if this 'learnt' information can be transferred into the real task scenario. During the real task, this step was monitored by the experimenter who, after participants finished the real tyre change, recorded a number of bolts that were not fully tightened i.e. the experimenter could unscrew the bolt with hand. For each bolt that was not tighten up correctly, the error penalty of 5 seconds was added to the overall completion time. Fig 5 shows a number of 5 second penalties recorded in each group during the real task.

Our results show that in the RO group, nine participants were given error penalties at the end of the real task. This group of participants received twenty-one 5 seconds penalties altogether. In the NC group, twelve participants received error penalties at the end of the real task. The total amount of 5 seconds penalties for these participants was forty-six. In the ATV group, ten participants received error penalties at the end of the real task. They cumulatively acquired nineteen 5 seconds penalties.

Statistical comparison tests (Mann-Whitney U) were conducted and the results revealed that the difference in the amount of penalties received was statistically significantly different between the two VR training groups, ATV and NC groups ($t_{(26)}$ = 1.873, p = 0.036, one tailed). No statistically significant differences in the amount of received error penalties were observed between the ATV and RO groups (p = 0.962) or between the RO and NC groups (p = 0.081). These findings support the notion that the availability of augmented multisensory cues during VR training can improve learning outcomes in the real environment and has a potential to reduce error rates on real tasks.

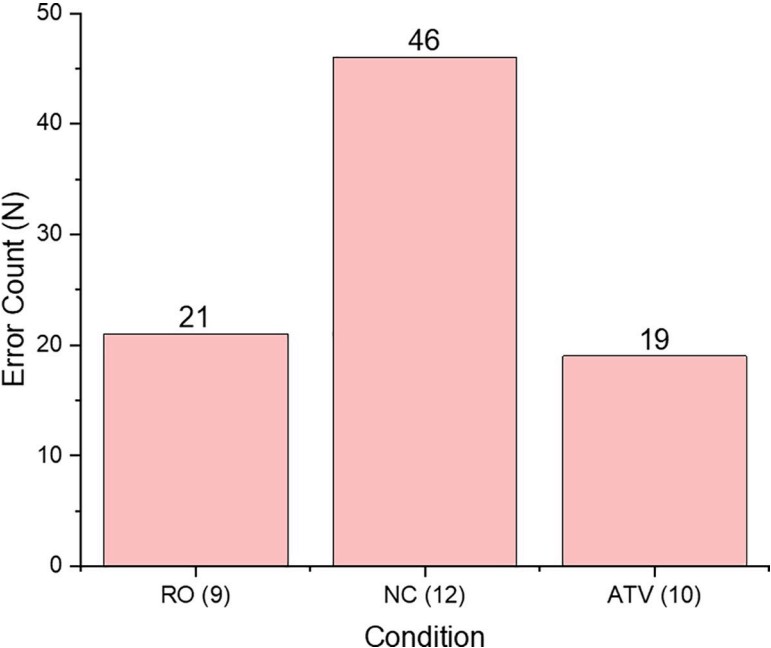

**Fig 5. Error penalties across groups.** The graph shows the amount of 5 seconds penalties recorded during the real task in each group (top of the bar). One error penalty (5 sec) was scored for each bolt that was not correctly tightened. Figures in brackets show the number of participants in that group who received error penalties—it was 9 participants in the RO group, 12 participants in the NC group, and 10 participants in the ATV group.

## Subjective measures

Before, during and after the task, all participants completed a number of questionnaires that were used as subjective measures: The NASA TLX workload questionnaire [38, 42] was used to record the perceived level of workload during the real and virtual task. In addition, participants who received virtual training also answered questionnaires probing immersive tendencies (ITQ) [36], perceived sense of presence (PQ) [36] and experienced discomfort (SSQ) [37].

**Perceived workload.** The levels of perceived workload were assessed with the NASA TLX workload questionnaire [38, 42]. Higher ratings on subscales indicate higher perceived demand, except for the performance subscale where a higher rating indicates *lower* self-perceived success on the task, this is indicated by arrows in Fig 6. The data were analysed by obtaining single workload score as well as by values on each workload subscales [42–44]. For a first analysis, the overall workload was computed as the average of all six subscales.

The data recorded during the real task shows that all groups experienced increased workload after the final trial run when compared to first trial run. This was expected as the task in this study was very physically demanding. A mixed ANOVA with training group (RO, ATV, NC) as between subject factor and workload subscales as a within participant factor shows significant main effects of both factors; $F_{(1,39)} = 9.797$, $p<0.001$, $n_2 = 0.33$) for training group, and $F_{(5,195)} = 12,723$. $p < 0.001$, $n_2 = 0.25$) for subscales. The interaction between two factors was shown to be not statistically significant ($p = 0.08$). Independent samples t-tests showed that the average workload ratings were significantly higher for the two groups that were trained in VR, compared to the group trained only on the real task (RO, M = 5.3, SD = 0.93); NC group (M = 6.72, SD = 0.98; $t_{(26)} = -3.956$, $p = 0.001$), ATV group (M = 6.53, SD = 0.86; $t_{(26)} = -3.643$, $p = 0.001$). There was no significant difference in overall perceived workload for two VR training groups when participants changed the real tyre ($t_{(26)} = 0.543$, $p = 0.592$). These findings are

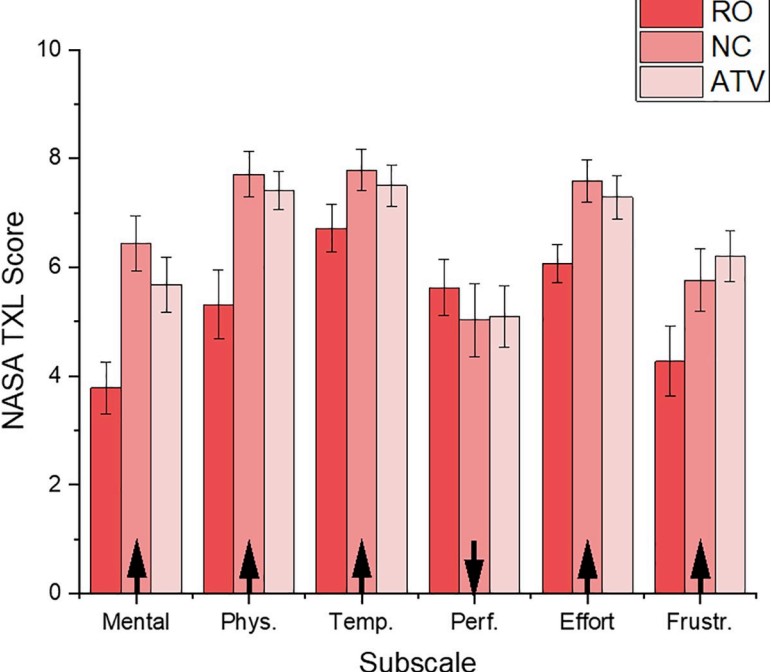

**Fig 6. Cognitive workload subscales.** Overall cognitive workload in all subscales across all experimental groups. Error bars represent SEM.

in line with previous research that shows higher workload ratings for VR trained groups compared with control groups [45–47]. When examining individual subscales, the NC group consistently reported slightly increased levels of workload during the real task for most subscales. Frustration subscale showed a slight change in ratings where the ATV group reported slightly higher rating than the NC group (ATV, M = 6.2, SD = 1.78; NC, M = 5.74, SD = 2.12). During virtual training, cognitive workload for both VR training groups was comparable. No significant differences were recorded between two VR training groups (NC, M = 5.22, SD = 1.36; ATV, M = 5.34, SD = 1.22; p = 0.343). This suggests that the presence of additional, augmented multisensory cueing during VR training did not impact workload levels in virtual environments.

**Perceived sense of presence.** Before the task, participants in the two VR training groups filled in the Immersive Tendencies Questionnaire (ITQ) [36]. No significant differences in immersive tendencies were found between the groups (NC: M = 102.2, SD = 23.2, ATV: M = 94.4, SD = 23.7; $t_{(28)}$ = 0.908, p = 0.372), which means that both groups had a same likelihood to become immersed in the virtual environment.

During the virtual training, the ratings for perceived sense of presence (PQ) [36] were also collected to determine how involved, immersed, and focused participants felt during the virtual training. Ratings were collected after the first and after the last trial. After the first trial, the NC group mean presence score was 193.6 (SD = 31.2) and the ATV group mean presence score was 188.2 (SD = 26.6). After the last trial, the ATV group mean presence score was 211.7 (SD = 33.4) and NC group mean presence score was 186.8 (SD = 41.7). A mixed ANOVA with 'training group' as the between-subject factor and PQ time-responses (first time, second time) as a within-subject factor revealed a significant interaction between groups and the time of the presence responses ($F_{(1,26)}$ = 6.139, p < 0.05, $n_2$ = 0.19). No other main effects showed significant results. Independent sample t-test revealed that the ATV group (M = 211,7; SD = 33.44) reported a significantly *higher* sense of presence at the end of the task when compared to the NC group (M = 186.75, SD = 41.7) ($t_{28}$ = 1.747, p = 0.04, one-tailed), which is consistent with our previous findings [7].

To examine which PQ subscale affected the overall perceived sense of presence on the final presence scores, a mixed ANOVA with subscales as a within subject factor and group as a between subject factor was conducted. Mean scores for each subscale can be seen in Fig 7. The analysis revealed a main effect of subscale ($F_{(3,78)}$ = 4.891, p < 0.01, $n^2$ = 0.16) and significant interaction between subscales and group ($F_{(3,78)}$ = 10.509, p < 0.001, n2 = 0.29). Independent sample t-tests showed that there was a significant difference on sensory subscale scores between the groups ($t_{28}$ = -4.053, p < 0.001). These findings show that the ATV group reported a significantly higher sense of presence, primarily based on the sensory subscale, than the NC group, which is also consistent with previous research [7].

**Simulation sickness and task performance.** As the task was performed in a virtual environment, participants also provided ratings of their experienced discomfort during the virtual training. The discomfort ratings were collected using the SSQ questionnaire [37] before participants were exposed to the VR (baseline), after the first trial, and at the end of the task, after trial 5. The mean weighed scores on each subscale relative to the baseline scores are shown in Fig 8.

A mixed ANOVA with group (NC, ATV) as a between-subjects factor and time (baseline, trial 1, trial 5) as a within-subject factor showed a significant main effect of time ($F_{(2,52)}$ = 7.02, p = 0.002), but no main effect of group. The scores recorded on all subscales reduced over time. Paired sample t-tests of the mean SSQ data, shown in Table 1, revealed that the SSQ scores reduced significantly relative to the baseline for the NC group (Trial 1: $t_{(26)}$ = 3.12, p = 0.008; Trial 5: $t_{(26)}$ = 2.87, p = 0.013). No significant changes over time were seen for the ATV group. The key finding of this analysis shows that performing the task in VR did *not increase* the SSQ scores relative to baseline during the training exercise.

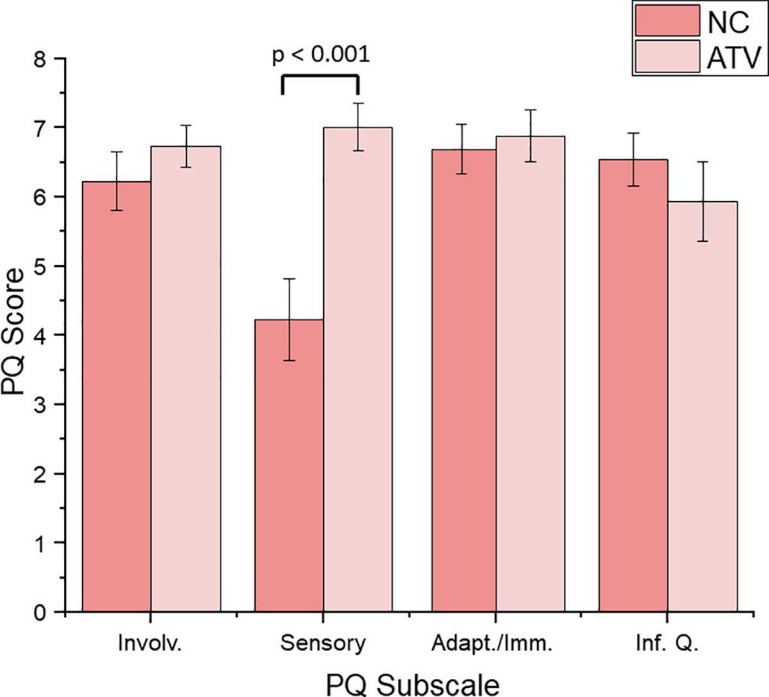

**Fig 7. Mean ratings for two VR training groups on PQ subscales.** Higher scores indicate a higher feeling of presence. The error bars represent SEM. Significant differences were observed on sensory subscale ($p < 0.001$).

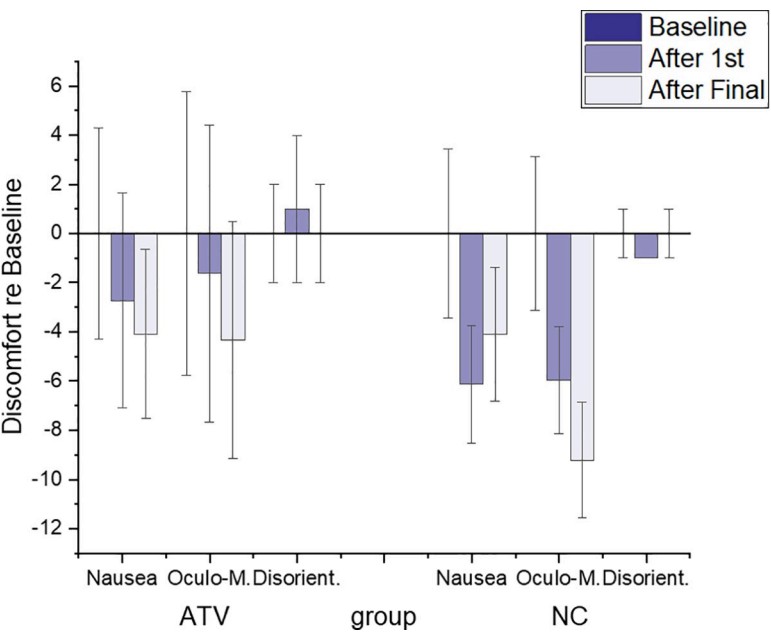

**Fig 8. Discomfort ratings.** Overall means of weighted scores for discomfort ratings obtained during the virtual training via the Simulation Sickness Questionnaire (SSQ). Error bars represent SEM.

**Table 1. Mean SSQ data for both VR training groups.**

| Group | Baseline M(SD) | Trial 1 M(SD) | Trial 5 M(SD) |
|---|---|---|---|
| ATV | 11.61 (13.51) | 10.49 (15.61) | 8.80 (11.50) |
| NC | 9.20 (7.65) | 4.84 (3.87) | 4.77 (5.21) |

## Discussion

Presenting task-relevant information using additional signals means that sensory fidelity is reduced because the presented sensory signals no longer objectively match real environments. On this basis one might expect a reduction of performance in VR as well as training transfer to real environments. These signals, of course, provide additional, task relevant information. We have previously shown that additional augmented multisensory cueing in VR not only enhances performance in the virtual environment, but, perhaps counterintuitively, also enhances user experience and perceived sense of presence in VR [7].

### Transfer of training from virtual to real scenarios

The primary research question we address here is whether augmented multisensory cues improve training transfer from virtual to real scenarios even when the overall realism or surface fidelity of the VR training simulation, as a direct result of the additional signals, is decreased (by providing additional colour and increased intensity of vibration and sound). In this study, we show that our strategy enhanced informational cue content within the VR simulation by adding arbitrary and unrealistic multisensory cues at the expense of overall realism. The former may lead to better learning transfer while the latter might be expected to reduce performance. Our results clearly show that VR training with these augmented cues has a beneficial effect not only on performance and user experience in the virtual environment, but also on real task performance. Both VR training groups performed better on the real task than the group that received no prior training in VR (RO group). Our results are in line with previous research that reported beneficial effects of virtual training [2–4]. During the real task, the gains for the group trained in virtual environment with augmented cues (ATV group) are higher than those for the conventional VR training group (NC group, as shown on Fig 4). This confirms our hypothesis that augmented, informationally enriched, multisensory cues enhance learning outcomes and transfer to a real environment, even though these cues reduce overall fidelity of the virtual environment during the training phase.

Performance measures obtained during learning tasks typically follow an exponential decline with rapid gains at the onset of training and relatively lower performance gains as time progresses [40]. This makes principled performance comparisons that are based on absolute performance measures difficult. Significant performance differences at the start of learning, as seen in our data, are typical for learning experiments. These differences reduce over time; in our experiments all three groups reach comparable performance levels after five trials. We propose an alternative measure that fits an exponential decay model with shared parameters to the data that quantifies the relative time shift (the equivalent training time) that training in VR causes.

Several studies have shown that an increased simulation fidelity supports better performance and learning outcomes [25, 27, 48, 49]. Designing VR systems to achieve the best possible fidelity is therefore an obvious design choice. In many situations high rendering quality, however, is not easily achievable. For this reason, the primary emphasis in the development of simulation-based training systems should be to maximize the effectiveness, not necessarily the fidelity [50, 51]. A number of factors, apart from fidelity, have been shown to affect training

transfer from VR to real environments. Below we outline how additional, augmented multisensory cues can potentially enhance learning outcomes in VR:

- Presence: The subjective feeling of 'being present' in a virtual world [15, 36]. The general assumption is that presence is enhanced with increasing fidelity and interactivity of the virtual environment [52]. In our previous work we have shown that cues that decrease the overall surface fidelity of the simulation, perhaps counterintuitively, enhanced subjective presence ratings, presumably because the augmented cues added functional fidelity [7].

- Attention [53]: Attention enhances learning outcome and can, of course, in VR be directed to arbitrary aspects in the simulation by additional cues.

- Overall learning strategy: Parong and Mayer [54] have shown that generative learning schemes, encouraging learners to actively engage with the material, improves performance; augmented cues in the virtual environment can control learner engagement with the simulation.

- Retention of learning outcomes: Carlson et al. [55] showed that colour-coding parts in an assembly task, which is relatively trivial in VR, but not an option in real assembly tasks, led to much better, and transferrable, retention of assembly procedures. We therefore argue that the use of augmented multisensory cueing for learning retention over time is worth investigating.

- Motivation: increase in intrinsic motivation (and enjoyment) has been linked to better transfer of learning in a physical (lab) environment [56]. Improving user experience and outcomes, which augmented cuing provides while reducing task load, is one way to improve learning motivation.

- Simulation variability: Externally controlled perturbations to the VR environments, within reasonable limits, has been shown to enable learners to acquire redundant approaches to problem solving [57, 58]. Additional augmented multisensory cues in VR may provide these controlled perturbations.

Error amplification: Errors that occur during training in virtual environments can be 'amplified' (in a stochastic or deterministic fashion) to improve performance [59, 60]. One of the invaluable advantages that VR has over real learning environments is that individual learner performance can be automatically monitored and feedback can be immediately provided.Our results show that the VR trained group that received augmented cueing information (ATV group) made significantly fewer errors on the real task than the group that trained in VR with basic visual information only (NC group, as shown on Fig 5). The ATV group received the fewest penalties at the end of the real task which means that most of the bolts were secured correctly. We can therefore conclude that the information provided in VR environment by augmented cues that signaled when the bolt was secured in correctly, was retained and participants were able to subsequently apply this knowledge in the real environment. This finding is of particular importance because it shows that the information fidelity enhancement provided at critical time points during VR training is a viable approach for the early detection and subsequent reduction of human errors in subsequent real tasks [1, 37, 61, 62]. In this study, we extend previous work that shows a link between simulation fidelity and training outcome [47, 49, 52], by further showing that additional, augmented multisensory cues used during virtual training that reduce surface fidelity, but carry task-relevant information, can, in fact, support training transfer and improve performance on real task scenarios.

## Perceived workload for effective task performance

The results for cognitive workload during the real task show that the perceived workload was lower at the beginning of the task, but increased systematically as participants completed more wheel change runs. During the virtual training both VR training groups reported comparable levels of workload. Previous research showed that multisensory cuing as well the immersive properties of the VR environment can contribute to increased cognitive workload [25, 43, 61, 63]. As both VR training groups reported high levels of presence during virtual training, with high levels of involvement and immersion, it is therefore plausible that these factors also contributed to their perceived workload levels.

During the real task, significant differences were observed between experimental groups: the RO group reported significantly lower cognitive workload when compared to both VR training groups, which suggest that VR training groups perceived the real task as more frustrating and more demanding (higher ratings on mental, physical and temporal workload subscales). These differences could be due to several factors.

One explanation for consistently higher workload ratings for VR training groups could be due to sampling effect. Previous work showed that perceived workload is not wholly defined by the task conditions, but is also sensitive to individual differences [64]. Other studies have suggested possible priming effects, for example prior exposure to VR training (i.e. expectations, fatigue) can influence overall perceived workload [37]. Changing the wheel in the virtual environment is much easier than changing the wheel in the real world, mainly due to absence of weight and torque forces experienced during the real tyre change task. As the RO group never experienced how 'easy' the virtual task was, they had nothing to compare the real task against. Although the weight of the impact wrench used in both tasks was comparable, the weight of the wheel during VR training was very different—zero weight during VR training compare to actual weight of the real tyre (approximately 7 kg each) during the real task. The experimenter noted that participants from VR training groups frequently commented on difference in overall physical workload between real and virtual task.

The overall cognitive workload for VR training groups (ATV and NC) during the real task was comparable as no significant difference between the groups was observed. Slightly higher levels of workload on individual subscale for the NC group could be a result of the amount of error penalties received during the real task, as NC groups received significantly more penalties than other groups. In a similar way, higher levels of frustration in the ATV group could be explained by the ATV group receiving most error penalties during the last run, which contributed to participant's frustration levels.

Our results further affirm the necessity of investigating how cognitive workload can contribute to the overall task completion, not only in virtual but also in real environments. Follow-up studies should concentrate on further evaluation and investigation of the effects of cognitive workload during VR training with multisensory cuing and the interactions between these variables. De-coupling the effects of individual sensory cues and their possible additional informational content on task performance can shed more light on how these features could further support transfer of training. The addition of objective workload measures would further enhance our understanding of how users' performance and acceptance of virtual environments can contribute to the effectiveness of training transfer to real environments.

## Augmented multisensory cues enhance user experience during virtual training

The analysis of subjective PQ data showed that the ATV group reported significantly higher overall presence than the group that receiving no additional cues and that this difference was

due to differences in *sensory* ratings. Although these survey questions do not directly test fidelity, they cover items such as the degree to which sensory cues involve the user, or whether users were able to examine objects. Additional cues that add to informational fidelity would be expected to improve these ratings. These findings are in line with previous research [7, 52, 65, 66] and further reinforce the point that informational fidelity, rather than purely the fidelity of sensory representations, has a direct effect on the perceived sense of presence as well as performance [7, 55, 67].

Our discomfort questionnaire data showed that as participants spend more time in VR, their overall discomfort *decreased* for both VR training groups, relative to baseline data that were captured before participants experienced the VR environment. Simulator sickness is often associated with imperfect participant motion tracking or visual signal presentation [37, 68]. In this case, increased SSQ scores relative to the baseline would be expected. The data therefore suggests that the high-quality VR system we used did not cause simulator sickness, which is in line with previous research that showed negative correlation between presence and cyber-sickness [69]. The SSQ scores for the ATV and NC training groups were not significantly different. This is consistent with the view that the augmented multisensory cues did not negatively impact discomfort despite the obvious reduction in simulation fidelity that the additional cues entail [69].

As any other research study, in this study some limitations were identified. One of these was our sample size as well as sample type. All of our participants were young university students and although we balanced our sample across gender and age, as well as considered individual difference, wider variations of our sample in these variable would provide more applicable results. The overall structure of the task could also be improved. For this study, we chose a wheel change task where performance was measured five times. We observed significant performance increases over the training period for all three groups, which were most obvious in the first training trials, while performance in the final trials were not significantly different between the groups. A more complex task, requiring more repetition to master the task, might have shown more subtle performance differences. Future studies could consider to design a task that is more complex and more challenging with a variation of subjects in terms of age, expertise and familiarity with VR. The addition of more precise objective measures that can track error measurements in more detail are also recommended, as well as using an additional control group to monitor for possible placebo effect of being trained. As the development of VR technologies is ongoing, forthcoming research could explore how advanced VR and AR technologies can be applied to further support effective learning in VR and enable successful training transfer to real environments.

## Conclusion

In order to support training and performance in the VR environment it is necessary to provide appropriate, task relevant, sensory and informational cues. Cooper et al. study [7] previously showed that increasing informational content, even if it disrupts fidelity, enhances performance and overall user experience in virtual environments. Here, we further show that additional informational content encapsulated in task relevant sensory cueing during virtual training can further enhance transfer of training from virtual to real life scenarios. One of the multisensory cues provided in this study (vibration) was chosen to represent cues that are difficult to simulate in VR environments (torque and weight, accompanied with additional visual and audio cues). We propose that task relevant and information-bearing cues can be an effective and efficient way to represent cues that are very hard to achieve in a VR environment (torque and weight) and we strongly encourage their use during virtual training in future research.

## Supporting information

**S1 Questionnaire. Immersive Tendencies Questionnaire (ITQ).**
(PDF)

**S2 Questionnaire. Presence Questionnaire (PQ).**
(PDF)

**S3 Questionnaire. Simulation Sickness Questionnaire (SSQ).**
(PDF)

**S1 File. NASA TLX workload scale.**
(PDF)

**S1 Data.**
(XLSX)

## Acknowledgments

The authors would like to thank other team members at the Virtual Engineering Centre at the University of Liverpool for providing technical support during the data collection phase.

## Author Contributions

**Conceptualization:** Natalia Cooper, Ferdinando Millela, Iain Cant, Mark D. White, Georg Meyer.

**Data curation:** Natalia Cooper, Ferdinando Millela, Iain Cant.

**Formal analysis:** Natalia Cooper, Ferdinando Millela, Georg Meyer.

**Funding acquisition:** Mark D. White, Georg Meyer.

**Investigation:** Natalia Cooper.

**Methodology:** Natalia Cooper, Georg Meyer.

**Project administration:** Natalia Cooper, Mark D. White, Georg Meyer.

**Resources:** Natalia Cooper, Iain Cant, Mark D. White, Georg Meyer.

**Software:** Ferdinando Millela, Iain Cant.

**Supervision:** Natalia Cooper, Ferdinando Millela, Mark D. White, Georg Meyer.

**Validation:** Natalia Cooper, Ferdinando Millela, Georg Meyer.

**Visualization:** Ferdinando Millela, Iain Cant.

**Writing – original draft:** Natalia Cooper, Georg Meyer.

**Writing – review & editing:** Natalia Cooper, Iain Cant, Mark D. White, Georg Meyer.

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
