## [Decision Letter · Decision Letter 0]

18 Dec 2020

PONE-D-20-29890

Transfer of training – virtual reality training with augmented multisensory cues improves user experience during training and task performance in the real world

PLOS ONE

Dear Dr. Cooper,

Thank you for submitting your manuscript to PLOS ONE. After careful consideration, we feel that it has merit but does not fully meet PLOS ONE’s publication criteria as it currently stands. Therefore, we invite you to submit a revised version of the manuscript that addresses the points raised during the review process.

The reviewers were positive about the contribution and have only minor comments. In contrast, I have concerns about several of the claims being only weakly or not supported by statistical inference, and thus the manuscript requires several changes to soften or remove some of the claims, including but not limited to those referring to P values at or above 0.05. Specifically:

235, 284 - p=0.05 should not be described as significant, but I suspect these values are in fact slightly above or below 0.05, in which case please provide exact P values.

237 - No effect on the last trial between the two groups? Doesn’t this at least partly undermine a principal claim of the paper? Even if other results support the hypothesis, it seems it would also predict a difference here, and thus not observing it deserves more than a perfunctory mention.

236, 285, 310 - More importantly, best practices suggest it is misleading and inappropriate to refer to 0.1 > p > 0.05 as a “trend towards significance.” One way to appreciate this is to consider p values less than but close to 0.05, which by the same logic indicate a “trend toward non-significance.” Some recent views on this are here:

https://www.bmj.com/content/348/bmj.g2215

https://academic.oup.com/bja/article/115/3/337/312358

https://www.ncbi.nlm.nih.gov/pmc/articles/PMC6440716/

It is highly preferred to use phrases like “failed to reach significance” or simply “not significantly different” for such results, and any conclusions made based on them (including in Discussion) should be removed, or else described as a possibility that must be evaluated in future research.

Lastly, here is a list of minor fixes and suggestions on the writing. Note that PLOS ONE does not copyedit manuscripts, and so you’ll be stuck with any typos that make it past the review stage:

52 - add ‘but’ after comma

122 - the ‘control’ group does not control for the possible placebo effect of having been trained. This is not a major problem but could be worth adding a caveat to the Discussion.

133 - add ‘a’ or ‘the’ after “in”

235 - “that RO group” -> than the RO group

232, 236, 258 - “trail” -> trial

275 - “a” -> the

275 -  “the bolts was not fully tighten” -> a bolt was not fully tightened

378 - “reduces” -> is reduced

397-399 - Please refer to which figure illustrates this result, and confirm its statistical significance per comments above.

447-448 - Same as above: indicate figure and confirm significance and not “trend towards”

525 - “were” -> was

We look forward to receiving your revised manuscript.

Kind regards,

Christopher R. Fetsch

Academic Editor

PLOS ONE

Journal Requirements:

2. Please change "female” or "male" to "woman” or "man" as appropriate, when used as a noun.

Reviewers' comments:

Reviewer's Responses to Questions

**Comments to the Author**

1. Is the manuscript technically sound, and do the data support the conclusions?

Reviewer #1: Yes

Reviewer #2: Yes

2. Has the statistical analysis been performed appropriately and rigorously? 

Reviewer #1: Yes

Reviewer #2: Yes

3. Have the authors made all data underlying the findings in their manuscript fully available?

Reviewer #1: Yes

Reviewer #2: Yes

4. Is the manuscript presented in an intelligible fashion and written in standard English?

Reviewer #1: Yes

Reviewer #2: Yes

5. Review Comments to the Author

Reviewer #1: The authors present a novel method to quantify the relative performance gains of students in virtual reality-enhanced learning environments. Specifically, the authors estimate differences between traditional training methods, traditional virtual reality training, and enhanced virtual reality environment.

The objective of the research as well as the methods are well explained. Also, the measurement instruments were well selected and led to meaningful results.

The discussion is well elaborated and accurately represents the research process. I consider that the manuscript does not need further adjustments.

Reviewer #2: Thank you very much for the opportunity to review this interesting manuscript which is an ongoing exploration of the addition of augmented multi sensory cues in the virtual environment and their effects on training and transfer of training into the real world environment. The authors conducted a study with 3 matched groups of participants assigned to "real world" tire change exercise, VR without augmented multi sensory cues training, or VR with augmented sensory cues training. I believe all participants performed 5 receptions of the "racing tire change" task with appropriate baseline assessments to confirm equivalence between groups.

Overall, the manuscript is well written and I feel can be accepted with minor revisions which I will outline below.

1. Introduction - well written and provides appropriate background for the reader to understand the material.

1.1. Ln 74 - I think you are missing "load" as part of "reduce cognitive".

2. Methods

2.1. Can the authors please provide a "study flow diagram" to explain graphically "baseline testing", number of repetitions performed in each group, testing performed after 1st attempt in each group, as well as clarify if and how many attempts participants in VR groups were allowed to make on the "real tire change" after training in the VR? I am still not clear on this point. Did participants in the VR groups perform 5 attempts in the VR and then 1 more attempt on "real tire change"?

2.2. Please provide an explanation or definition for what "immersive tendencies" and "perceived sense of presence" means in the VR environment. This will help the readers familiarize themselves with these concepts.

2.3. Did you perform a power calculation to help you decide how many participants were required for each study group? If yes, please include this in the manuscript.

2.4. Please move relevant descriptions of NASA TLX, ITQ, and SSQ questionnaires and their interpretation from Results section to Methods section.

3. Results

3.1. Ln 254 - 256 - can the authors please explain what this means? Does training in ATV VR is 80% as effective at training on "real life" model? Or, training in ATV VR is more effective than real life model. It is still confusing for me and I suspect will be for other readers.

3.2. Ln 265 - similar to my comment above, please explain this a bit more. Do you expect individuals to achieve the same "level of skill" or "outcome" by training 2/3 of the total time in ATV VR as compared to real life? Is it more effective?

4. Discussion

4.1. Ln 388-389 - can you please point me to the data to support the statement that realism of the VR training is decreased? I was not able to find this in the manuscript.

4.2. Ln 393 - 394 - can you please point me to the results that support the statement that both VR groups performed better on real life task than RO group? This goes back to my comment previously about not being able to understand if VR group participants actually performed greater number of total repetitions (VR + real life) than RO participants. If yes, this is a possible explanation for this finding that should be discussed.

4.3. Ln 453 - can you please indicate where "surface fidelity" was measured and how?

4.4. Consider including a limitations section as one is not found in the manuscript.

4.5 Ln - 515 - can you please explain why addition of cues results in reduction of simulation fidelity? I would think that adding more "sounds", "haptics", etc. would increase fidelity as long as there is task alignment.

Boris Zevin MD, PhD

6. PLOS authors have the option to publish the peer review history of their article (what does this mean?). If published, this will include your full peer review and any attached files.

Reviewer #1: **Yes: **Juan Garzón

Reviewer #2: **Yes: **Boris Zevin MD, PhD

---

## [Author Response · Author response to Decision Letter 0]

3 Feb 2021

Dear Dr Fetch 

 Thank you – and the two reviewers – for taking the time to review the manuscript and to offer positive and constructive feedback.

We kept your comments from the decision letter and added our response in green to show how the comments were addressed. 

Thank you for submitting your manuscript to PLOS ONE. After careful consideration, we feel that it has merit but does not fully meet PLOS ONE’s publication criteria as it currently stands. Therefore, we invite you to submit a revised version of the manuscript that addresses the points raised during the review process.

The reviewers were positive about the contribution and have only minor comments. In contrast, I have concerns about several of the claims being only weakly or not supported by statistical inference, and thus the manuscript requires several changes to soften or remove some of the claims, including but not limited to those referring to P values at or above 0.05. Specifically:

235, 284 - p=0.05 should not be described as significant, but I suspect these values are in fact slightly above or below 0.05, in which case please provide exact P values.

These have been amended as suggested. 

237 - No effect on the last trial between the two groups? Doesn’t this at least partly undermine a principal claim of the paper? Even if other results support the hypothesis, it seems it would also predict a difference here, and thus not observing it deserves more than a perfunctory mention.

We disagree with the claim that not finding performance differences at the end of (the real) training undermines the principal claim. The key point to emphasize is that these trials are performance measures on the same (real) task after participants were trained in VR. The "study flow diagram" requested by reviewer 2 probably will help here. 

Learning curves are exponential functions, their asymptotic behaviour means that we expect all participant groups to – eventually – arrive at comparable performance. In our experiment that is relatively fast, after 2 tyre changes we see little performance improvement in any of the three groups. The key argument we make is that VR training (and especially VR training with augmented cues) provides participants with a ‘head start’ when performing task in the real world. We do not expect to see significant performance differences towards the end of the training on the real task 

236, 285, 310 - More importantly, best practices suggest it is misleading and inappropriate to refer to 0.1 > p > 0.05 as a “trend towards significance.” One way to appreciate this is to consider p values less than but close to 0.05, which by the same logic indicate a “trend toward non-significance.” Some recent views on this are here:

https://www.bmj.com/content/348/bmj.g2215

https://academic.oup.com/bja/article/115/3/337/312358

https://www.ncbi.nlm.nih.gov/pmc/articles/PMC6440716/

It is highly preferred to use phrases like “failed to reach significance” or simply “not significantly different” for such results, and any conclusions made based on them (including in Discussion) should be removed, or else described as a possibility that must be evaluated in future research.

The new sections have been added and the text has been updated accordingly. 

Lastly, here is a list of minor fixes and suggestions on the writing. Note that PLOS ONE does not copyedit manuscripts, and so you’ll be stuck with any typos that make it past the review stage:

52 - add ‘but’ after comma

- done

122 - the ‘control’ group does not control for the possible placebo effect of having been trained. This is not a major problem but could be worth adding a caveat to the Discussion.

The text addressing this was added to the Discussion section.

133 - add ‘a’ or ‘the’ after “in” - done

235 - “that RO group” -> than the RO group - done

232, 236, 258 - “trail” -> trial – all replaced

275 - “a” -> the -done

275 - “the bolts was not fully tighten” -> a bolt was not fully tightened – replaced with ‘when bolts were not fully tightened’

378 - “reduces” -> is reduced -done

397-399 - Please refer to which figure illustrates this result, and confirm its statistical significance per comments above.

The text in this section was amended as suggested.

447-448 - Same as above: indicate figure and confirm significance and not “trend towards”

The text in this section was amended as suggested

525 - “were” -> was - done

We look forward to receiving your revised manuscript.

5. Review Comments to the Author

Reviewer #1: The authors present a novel method to quantify the relative performance gains of students in virtual reality-enhanced learning environments. Specifically, the authors estimate differences between traditional training methods, traditional virtual reality training, and enhanced virtual reality environment.

The objective of the research as well as the methods are well explained. Also, the measurement instruments were well selected and led to meaningful results.

The discussion is well elaborated and accurately represents the research process. I consider that the manuscript does not need further adjustments.

Reviewer #2: Thank you very much for the opportunity to review this interesting manuscript which is an ongoing exploration of the addition of augmented multi sensory cues in the virtual environment and their effects on training and transfer of training into the real world environment. The authors conducted a study with 3 matched groups of participants assigned to "real world" tire change exercise, VR without augmented multi sensory cues training, or VR with augmented sensory cues training. I believe all participants performed 5 receptions of the "racing tire change" task with appropriate baseline assessments to confirm equivalence between groups.

Overall, the manuscript is well written and I feel can be accepted with minor revisions which I will outline below.

1. Introduction - well written and provides appropriate background for the reader to understand the material.

1.1. Ln 74 - I think you are missing "load" as part of "reduce cognitive". -done

2. Methods

2.1. Can the authors please provide a "study flow diagram" to explain graphically "baseline testing", number of repetitions performed in each group, testing performed after 1st attempt in each group, as well as clarify if and how many attempts participants in VR groups were allowed to make on the "real tire change" after training in the VR? I am still not clear on this point. Did participants in the VR groups perform 5 attempts in the VR and then 1 more attempt on "real tire change"?

The text to explain this further was added as well as the figure showing study flow diagram.

2.2. Please provide an explanation or definition for what "immersive tendencies" and "perceived sense of presence" means in the VR environment. This will help the readers familiarize themselves with these concepts.

We expanded the introductory section (L69-79) to introduce both concepts:

It is well understood that performance in VR training environments is directly affected by simulation fidelity [11–13]. Poor mechanical performance of simulated haptic feedback, for example, has been shown to result in negative training outcomes in surgical training [14]. Conversely, improved simulation fidelity not only captures the sensory cues and behaviours that learners encounter in real life better, but it also improves presence and immersion in VR environments [11–13,15–17] and reduce cognitive load [18–20]. ‘Immersion’ here stands for the objective level of sensory fidelity provided by a VR system, which is a direct measure of simulation fidelity. ‘Presence’ describes the subjective psychological response of a user experiencing a VR system as ‘being in the virtual world’ (see [7] for further discussion of these terms). For successful transfer of training from virtual into real environments, it therefore seems essential that the cues presented during the training in VR are representative of those encountered in reality.

2.3. Did you perform a power calculation to help you decide how many participants were required for each study group? If yes, please include this in the manuscript.

We did not perform a formal power analysis when planning the experiment: The main research question was whether VR training transfers to a real task, we would have had to assume an effect size. This experiments reported here, however, are a continuation of Cooper et al (2018) where we found highly significant differences in subjective and objective measures (effect sizes Cohen’s d=3.8, objective, and d=5.63, subjective) when comparing the two VR conditions chosen for the experiments reported in this study. A post-hoc power analysis (g-power, diff between independent means, alpha and beta =0.05, allocation ration=1) suggests that we would have ‘got away’ with just six participants (actual power 0.982) if our aim was to show the differences between augmented and ‘normal’ VR training. This is in line with data presented here [Cooper et al 2018 https://doi.org/10.1371/journal.pone.0191846] 

2.4. Please move relevant descriptions of NASA TLX, ITQ, and SSQ questionnaires and their interpretation from Results section to Methods section.

This section has been amended.

3. Results

3.1. Ln 254 - 256 - can the authors please explain what this means? Does training in ATV VR is 80% as effective at training on "real life" model? Or, training in ATV VR is more effective than real life model. It is still confusing for me and I suspect will be for other readers.

The section was added to the text for better explanation.

The learning curve, also known as experience curve, represents the improved efficiency or performance obtained from repeating an operation of a specific task by a worker; i.e. the time required to perform a task declines at a decreasing rate as experience with the task increases [33].

To enable a direct comparison of the data the three curves were fitted simultaneously. Parameters A1, x¬1 and y0 were shared across the three data sets under the assumption that, since each of the three (matched) groups performed the same task, we assume initial and final performance as well as learning rate to be the same for all three groups. The x-offset (x0), in contrast, was fitted for each individual to quantify the equivalent training time (number of trial runs) that was gained by participating in the VR training exercise preceding the real tyre changes. The difference in x0¬ between the ‘real only’ (RO) condition and the two VR training conditions therefore represents the ‘real training time saved’ by taking part in the initial VR training.

3.2. Ln 265 - similar to my comment above, please explain this a bit more. Do you expect individuals to achieve the same "level of skill" or "outcome" by training 2/3 of the total time in ATV VR as compared to real life? Is it more effective?

The section has been added to the text to provide a better explanation.

To estimate the shift parameter (x0), a bootstrapping analysis with 10000 resamples (with replacement) was carried out to estimate the mean x-offset (x0) and 95% confidence intervals for each of the three conditions. The mean x-offset ((x0) ®) for the RO group was 1.10 (CI 0.51 - 1.63), for the NC group (x_0 ) ® = 0.554 (CI -0.0423 - 1.171) while for the ATV group (x_0 ) ® = 0.1735 (CI -0.3903 - 0.6853). The probability that the observed mean x-offset values were drawn from the RO data are p=0.0012 (ATV) and p=0.0329 (NC) respectively. The relative differences in the x0 parameter mean that conventional VR training (NC) is equivalent to 0.37 ‘training sessions’ on a real tyre while the augmented (ATV) VR training provides gains that are equivalent to 0.93 real tyre changes (Fig 3).

4. Discussion

4.1. Ln 388-389 - can you please point me to the data to support the statement that realism of the VR training is decreased? I was not able to find this in the manuscript.

A more detailed explanation is presented in the text. Line 107-119 also explain this.

4.2. Ln 393 - 394 - can you please point me to the results that support the statement that both VR groups performed better on real life task than RO group? This goes back to my comment previously about not being able to understand if VR group participants actually performed greater number of total repetitions (VR + real life) than RO participants. If yes, this is a possible explanation for this finding that should be discussed.

The text has been added as well as study flow diagram.

4.3. Ln 453 - can you please indicate where "surface fidelity" was measured and how?

Neither ‘surface fidelity’, nor ‘informational fidelity’, the two terms we use in the context, were measured. However, there is a well-rehearsed argument that an “exact measure of realism is not feasible at this time, and is considered by some to be “a goal which can never be accomplished” (Roza et al., 2001).” (from Liu et al., Liu, D., Macchiarella, N. D., & Vincenzi, D. A. (2008). Simulation fidelity. Human factors in simulation and training, 61-73.) 

To address this comment we expanded our text to clearly define the terms we use from the outset. The key point we make is that we deliberately introduce signals, which cannot occur in reality, the colour-change when a tyre is correctly seated, for examples. These changes, by definition, reduce ‘surface fidelity’. 

4.4. Consider including a limitations section as one is not found in the manuscript.

This section has been added. 

4.5 Ln - 515 - can you please explain why addition of cues results in reduction of simulation fidelity? I would think that adding more "sounds", "haptics", etc. would increase fidelity as long as there is task alignment.

A section explaining this has been added to the introduction because it is a key consideration for the experiments.

Boris Zevin MD, PhD

6. PLOS authors have the option to publish the peer review history of their article (what does this mean?). If published, this will include your full peer review and any attached files.

Do you want your identity to be public for this peer review? For information about this choice, including consent withdrawal, please see our Privacy Policy.

Reviewer #1: Yes: Juan Garzón

Reviewer #2: Yes: Boris Zevin MD, PhD

---

## [Decision Letter · Decision Letter 1]

23 Feb 2021

Transfer of training – virtual reality training with augmented multisensory cues improves user experience during training and task performance in the real world

PONE-D-20-29890R1

Dear Dr. Meyer,

We’re pleased to inform you that your manuscript has been judged scientifically suitable for publication and will be formally accepted for publication once it meets all outstanding technical requirements.

Kind regards,

Christopher R. Fetsch

Academic Editor

PLOS ONE

Additional Editor Comments (optional):

Reviewers' comments:

Reviewer's Responses to Questions

**Comments to the Author**

1. If the authors have adequately addressed your comments raised in a previous round of review and you feel that this manuscript is now acceptable for publication, you may indicate that here to bypass the “Comments to the Author” section, enter your conflict of interest statement in the “Confidential to Editor” section, and submit your "Accept" recommendation.

Reviewer #2: All comments have been addressed

2. Is the manuscript technically sound, and do the data support the conclusions?

Reviewer #2: Yes

3. Has the statistical analysis been performed appropriately and rigorously? 

Reviewer #2: Yes

4. Have the authors made all data underlying the findings in their manuscript fully available?

Reviewer #2: Yes

5. Is the manuscript presented in an intelligible fashion and written in standard English?

Reviewer #2: Yes

6. Review Comments to the Author

Reviewer #2: Thank you very much for addressing my previous comments. I believe the revised version of the manuscript is much improved and appropriate for acceptance.

7. PLOS authors have the option to publish the peer review history of their article (what does this mean?). If published, this will include your full peer review and any attached files.

Reviewer #2: **Yes: **Boris Zevin

---

## [Editor Report · Acceptance letter]

26 Feb 2021

PONE-D-20-29890R1 

Transfer of training – virtual reality training with augmented multisensory cues improves user experience during training and task performance in the real world 

Dear Dr. Meyer:

I'm pleased to inform you that your manuscript has been deemed suitable for publication in PLOS ONE. Congratulations! Your manuscript is now with our production department. 

Kind regards, 

on behalf of

Dr. Christopher R. Fetsch 

Academic Editor

PLOS ONE